# Medical Device-Associated Healthcare Infections: Sterilization and the Potential of Novel Biological Approaches to Ensure Patient Safety

**DOI:** 10.3390/ijms25010201

**Published:** 2023-12-22

**Authors:** Mary Garvey

**Affiliations:** 1Department of Life Science, Atlantic Technological University, F91 YW50 Sligo, Ireland; mary.garvey@atu.ie; 2Centre for Precision Engineering, Materials and Manufacturing Research (PEM), Atlantic Technological University, F91 YW50 Sligo, Ireland

**Keywords:** ESKAPE, fungal, pathogen, HAI, priority, medical device, sterilization

## Abstract

Healthcare-associated infections caused by multi-drug-resistant pathogens are increasing globally, and current antimicrobial options have limited efficacy against these robust species. The WHO details the critically important bacterial and fungal species that are often associated with medical device HAIs. The effective sterilization of medical devices plays a key role in preventing infectious disease morbidity and mortality. A lack of adherence to protocol and limitations associated with each sterilization modality, however, allows for the incidence of disease. Furthermore, issues relating to carcinogenic emissions from ethylene oxide gas (EtO) have motivated the EPA to propose limiting EtO use or seeking alternative sterilization methods for medical devices. The Food and Drug Administration supports the sterilization of healthcare products using low-temperature VH_2_O_2_ as an alternative to EtO. With advances in biomaterial and medical devices and the increasing use of combination products, current sterilization modalities are becoming limited. Novel approaches to disinfection and sterilization of medical devices, biomaterials, and therapeutics are warranted to safeguard public health. Bacteriophages, endolysins, and antimicrobial peptides are considered promising options for the prophylactic and meta-phylactic control of infectious diseases. This timely review discusses the application of these biologics as antimicrobial agents against critically important WHO pathogens, including ESKAPE bacterial species.

## 1. Introduction

Healthcare-associated infections (HAI) or nosocomial infections are a major public health risk, with hospitalized patients manifesting infectious symptoms ca. 48 h post-admission [1]. Importantly, studies show that symptoms can manifest post-discharge, as seen with ca. 60% of surgical-site infections (SSIs), which is associated with increased mortality [2]. HAIs associated with medical devices, including ventilator-associated pneumonia (VAP), central-line-associated bloodstream infections (CLABSIs), catheter-related urinary tract infection (CAUTI), and SSIs, lead to prolonged hospital stays, risk of sepsis, and mortality in intensive care units (ICUs) [3]. There is an increased risk of HAIs in ICU patients, with ca. 30% acquiring at least one infection and with significant subsequent morbidity and mortality [4]. The increasing rates of antimicrobial resistance (AMR) present in nosocomial pathogens proliferate the risks associated with HAIs [5]. Medical device-associated infections are more commonly associated with reusable medical devices, including surgical forceps, endoscopes, bronchoscopes, transesophageal echographs, and laryngoscopes, compared to single-use devices. In the United States, ca. 1.7 million cases of HAIs and 99,000 deaths occur annually, where medical devices are causative of ca. 80% of UTIs, BSIs, and pneumonia in admitted patients [6]. In Europe, the prevalence of HAIs is 7.1%, with ca. 4 million cases annually [4]. Reusable devices must be reprocessed or decontaminated (cleaning, disinfection, and/or sterilization) to eradicate microbial life prior to subsequent use. Single-use devices are terminally sterilized and less prone to contamination. Such devices are classified according to Spaulding’s classification system as critical devices that have direct contact with blood and or internal tissues, semi-critical devices that have contact with mucous membranes, and non-critical devices contacting intact skin [7].

Reusable devices must be reprocessed in accordance with the International Standard Organization (ISO) ISO 17664-1:2021 [8] for devices requiring cleaning, disinfection, and/or sterilization by clinical practices before reuse, as outlined by the Medical Device Regulation harmonization [8]. Importantly, the disinfection or reuse of implantable medical devices is not permitted, as it risks patient safety [9]. Critical medical devices require sterilization, semi-critical medical devices require high-level disinfection or sterilization, and non-critical devices require low- to intermediate-level disinfection [10]. HAIs associated with reusable medical devices are related to a failure to comply with these reprocessing guidelines [11,12]. Alarmingly, medical device HAIs associated with MDR pathogens have occurred where no identifiable non-compliance in reprocessing occurred [6]. The suitability of the Spaulding system, which is approximately 65 years old, for reprocessing novel, advanced, and complex medical devices to eliminate emerging and MDR pathogens, is questionable [13]. The FDA’s Center for Devices and Radiological Health (CDRH) and The Sterility and Infection Control Program (incorporating the sub-programs “Alternatives to Ethylene Oxide Sterilization” and “Device-Related Infections”) aim to ensure patient safety by applying regulatory scientific research to provide sterile medical devices as needed [14]. In an antibiotic-resistant era of emerging and re-emerging MDR pathogens, it is imperative to ensure patient safety. This opportune review highlights key factors associated with the incidence of HAIs and sterilization, as well as disinfection failure of reusable device microbial virulence factors enabling survival and persistence on devices. It also discusses the impact of WHO-priority pathogens on morbidity and the mortality rates of affected patients. The application of novel antimicrobial techniques is also discussed in terms of feasibility in clinical settings towards antimicrobial stewardship.

## 2. Microbial Pathogens Associated with AMR HAIs

Both the WHO bacterial and fungal priority pathogen lists detail critically and highly important pathogens displaying high levels of AMR where treatment protocols are increasingly difficult to determine [15]. The ESKAPE (*Enterococcus faecium*, *Staphylococcus aureus*, *Klebsiella pneumoniae*, *Acinetobacter baumannii*, *Pseudomonas aeruginosa*, and *Enterobacter* species) pathogens are nosocomial and associated with high levels of resistance and increased morbidity and mortality [15]. Clinical isolates of ESKAPE pathogens have proven to be MDR, extensively drug-resistant (XDR), and pan-drug-resistant (PDR) [16]. Similarly, many of the fungal priority pathogens, e.g., *Candida*, *Aspergillus*, and *Cryptococcus* spp., have frighteningly high mortality rates due to the robust and invasive nature and MDR of these species [7]. Microbial species possess many virulence factors, allowing them to survive, persist, proliferate, and transmit in clinical settings. Additionally, microbial efflux pumps in bacterial and fungal cells allow for combined therapeutic and biocidal resistance [5].

### 2.1. ESKAPE Pathogens Associated with Medical Device HAIs

The WHO’s critically important pathogens include Carbapenem-resistant *A. baumannii* and *P. aeruginosa*, β-lactamase (ESBL) or Carbapenem-resistant *K. pneumoniae*, and *Enterobacter* spp. (carbapenem-resistant Enterobacterale (CRE)), with vancomycin-resistant *E. faecium (VRE)*, methicillin-resistant *S. aureus* (MRSA), and vancomycin-resistant *S. aureus* listed as high-priority pathogens [17]. Small-colony variants of *S. aureus* have also emerged, which are causative of persistent difficult-to-treat dermal infections, with *S. aureus* bacteremia having a ca. 30% mortality rate [15]. The CDC reports that *A. baumannii* caused 8500 HAI cases and 700 deaths, ESBL Enterobacterale caused 197,400 cases and 9100 deaths, MDR *P. aeruginosa* caused 32,600 cases and 2700 deaths, with CRE causing 13,100 HAI cases and 1100 deaths in 2017 in the US [18]. Additionally, Gram-positive pathogens VRE resulted in 54,500 HAI cases and 5400 deaths, with MRSA causing 323,700 cases and 10,600 deaths in 2017 [18]. VRE shared the gene for vancomycin resistance with *S. aureus*, resulting in the emergence of vancomycin-resistant *S. aureus* (VRSA), thus limiting the application of this last-resort antibiotic against Gram-positive pathogens [16]. *A. baumannii* is associated with HAIs primarily related to respiratory equipment and indwelling catheters, resulting in pneumonia, sepsis, UTIs, endocarditis, and meningitis [19]. *A. baumannii* is a high-risk nosocomial pathogen due to its innate and acquired AMR, long-term survival on dry surfaces [20], and adaptability to varying ecological niches [21]. Carbapenems have been used to treat MDR Gram-negative species, including *A. baumannii*. However, Carbapenem-resistant strains that produce metallo-beta-lactamase (MBLs) enzymes have now emerged, reducing antibiotic efficacy [22]. Pan-drug-resistant *A. baumannii*, which possesses resistance to all antibiotics, excluding colistin and tigecycline, has been identified [23]. In 2014, carbapenem resistance *K. pneumoniae* and MDR *P. aeruginosa* were isolated from 19 infected patients undergoing bronchoscopy in the ICU [24]. Last-resort antibiotics tigecycline and colistin have been applied to treat Carbapenem-resistant strains. However, these therapeutics have biocompatibility relating to kidney and liver toxicity, with resistance also emerging in some species [25]. Colistin resistance in *A. baumannii* results from the loss of lipopolysaccharide (LPS) from the bacterial cell membrane [15]. There are increasing rates of acquisition of a plasmid-mediated imipenem-hydrolyzing enzyme (KPC-2) by numerous Carbapenem-resistant species, including *Klebsiella* [23]. Colistin-resistant *P. aeruginosa*, which is considered a pan-drug-resistant strain, poses a serious risk in clinical settings [26]. The enteric *Enterobacter* species are associated with MDR HAIs, with plasmid-encoded ESBLs and KPC carbapenemases, including OXA, metallo-β-lactamase, and metallo-β-lactamase-1 resistance mechanisms [23]. *Enterobacter* is associated with UTIs, SSIs, intravascular medical device infections, and bacteremia, with high mortality rates of ca. 25%, increasing to ca. 35% where cephalosporin-resistant strains were present [27]. Alarmingly, the repertoire of antibiotic and antibiotic combinations for treating these ESKAPE pathogens is decreasing yearly, with the Clinical & Laboratory Standards Institute (CLSI) detailing updated minimum inhibitory concentration (MIC) breakpoints with indicators of resistance [28]. The presence of persister cells and viable but non-culturable (VBNC) cells allows certain species to survive exposure to unfavorable conditions and evade detection methods, allowing for nosocomial transmission [21]. VBNC and persister cells also have increased AMR, increased antibiotic tolerance, increased stress tolerance, and active virulence factors promoting pathogenesis [29]. The bacterial LPS endotoxin released from Gram-negative species post-cell death is also an important consideration in the reprocessing of medical devices. As a pyrogen, the LPS toxin is associated with inflammation, sepsis, and mortality in patients and has been linked to cardiovascular failure and organ failure [5]. The International Standard ISO 21582:2021 [30]—Pyrogenicity details the principles and methods for pyrogen testing of medical devices and materials to safeguard patients [30].

### 2.2. Fungal Priority Pathogens Associated with Medical Device HAIs

Many fungal species, including *Candida albicans*, are commensal organisms of the human body found on the skin or in the oral cavity, respiratory tract, and gastrointestinal tract (GIT) [31]. Localized and systemic fungal infections typically involve opportunistic *Candida*, *Aspergillus*, *Mucorale*, and *Fusarium* species [32]. Invasive fungal infections are associated with increased rates of chronic morbidity and mortality with WHO critically important species *C. albicans*, *C. auris*, *Cryptococcus neoformans*, and *Aspergillus fumigatus* commonly isolated [33]. Non-albicans species of *Candida* listed as a high priority, including *C. glabrata*, *C. tropicalis*, and *C. parapsilosis*, with *Mucorales* and *Fusarium* species also prone to nosocomial transmission. The innate AMR of fungal species, lack of antifungal therapeutic options, poor diagnostic methods, drug biocompatibility, and absorption issues mean invasive fungal infections represent a significant challenge in clinical settings [32]. Fungal HAIs have been associated with prostheses, catheters, and mechanical heart devices, with established infections being hard to treat and often fatal [34]. Trends indicate that the prevalence of fungal endocarditis is increasing, with *C. albicans* and *Aspergillus* the most common fungal isolates [35]. Cases often involve implanted prosthetic cardiac devices, e.g., prosthetic valves [36]. In May 2023, the CDC reported an incidence of fungal meningitis following a surgical procedure, leading to two fatalities. *Fusarium solani* was identified as the causative agent [37]. Studies show that 80% of candidiasis patients have implanted medical devices [38]. *Candida* species are associated with ca. 90% of fungal BSIs, resulting in sepsis and with a mortality rate of ca. 40% with treatment [7]. *Aspergillus fumigatus* is the most common fungal species infecting the respiratory tract of immunocompromised patients, resulting in severe and fatal invasive infections [34]. Studies describe *C. neoformans* endocarditis in patients with implanted cardiac medical devices such as cardiac valves [35]. Certain fungal species, including *Aspergillus* and *Fusarium* species, release mycotoxins, which are potent virulence factors resulting in host toxicity. Many mycotoxins are lipophilic compounds, allowing for their absorption and distribution in vivo [39]. *Aspergillus* mycotoxins (aflatoxins, ochratoxins, gliotoxin, fumonisins, and patulin) have nephrotoxic, genotoxic, teratogenic, carcinogenic, and cytotoxic activity [40]. Mycotoxins are released as secondary metabolites and are significantly difficult to detect and identify as causative agents of illness.

### 2.3. Microbial Biofilms Are a Significant Factor in Pathogenesis

Biofilms are microbial communities enclosed in an extracellular polysaccharide matrix (EPS) formed attached to a surface with sufficient moisture present. The adhesive EPS matrix consists of proteins, fatty acids, and polysaccharides, which form water channels, allowing for water and oxygen distribution among resident cells [41]. The proximity of species within biofilms, particularly polymicrobial biofilms, allows for the exchange of plasmids encoding resistance genes among species via horizontal gene transfer (HGT) more efficiently than observed in planktonic or free-floating cells [42], thus proliferating the emergence of resistance phenotypes and MDR species. Sessile or biofilm cells have increased AMR and protection against antimicrobial agents, including active pharmaceutical ingredients (APIs), biocidal chemicals, and host immune systems in vivo [41]. The EPS limits penetration by APIs, biocides, antibodies, white blood cells, and additional host defenses [43]. Indeed, the treatment of biofilms in vivo often necessitates concentrations of antimicrobial therapeutics that are toxic to the patient [44]. The presence of biofilms on medical devices prevents healing and leads to BSIs from the release of planktonic cells, implant rejection, morbidity, and mortality in some cases [45]. Of the ESKAPE pathogens, the most associated with medical device biofilm infections are *E. faecalis*, *S. aureus*, *E. coli*, *K. pneumoniae*, and *P. aeruginosa* [41]. *Staphylococcus* species are believed to be associated with ca. 50% of prosthetic heart valve infections, 50–70% of catheter biofilm infections, and 87% of BSIs [46]. Indeed, biofilms are often present on devices including infusion pumps, oxygen machines, mechanical ventilators, [47] and endoscopes [12]. Biofilm-residing colistin-resistant *P. aeruginosa* was isolated from bronchoscopes [25]. Biocidal resistance in biofilms is of major concern, as studies show polymicrobial biofilms of *P. aeruginosa* and *K. pneumonia* have resistance to chlorhexidine and hydrogen peroxide (H_2_O_2_) at clinical concentrations [48,49]. Peracetic acid displays improved efficacy against *A. baumannii*, *K. pneumoniae*, and *P. aeruginosa* biofilms [43]. Studies describe the presence of polymicrobial MDR biofilms on medical equipment and furnishings in ICUs for ca. 12 months post-cleaning with antimicrobial solutions, e.g., bleach [50]. Studies also describe polymicrobial biofilms on urinary catheters resulting in UTIs with *Enterococcus faecalis*, *Escherichia coli*, and *Klebsiella pneumoniae* isolated in polymicrobial biofilms, and *Enterococcus* spp., *Pseudomonas aeruginosa*, and *Pseudomonas mirabilis* present in some patients [50].

Yeast and fungal species are also proficient biofilm formers. *Candida* sp., for example, has established biofilms on medical devices, including vascular catheters, resulting in BSIs and disseminated infection, resulting in mortality rates of ca. 30% [38]. *C. albicans* was identified in 45% of patients manifesting with respiratory tract infections resulting from long-term intubation [31]. *A. fumigatus* can form biofilms on biotic and abiotic surfaces and often results in a severe, difficult-to-treat, and fatal infection [34]. *Aspergillus* biofilms have been detected on catheters, prosthetic devices, pacemakers, heart valves, and breast implants [45]. Zhang et al. (2021) described the detection of fungi on 67% of peripherally inserted central vein catheters [51]. *C. neoformans* biofilms have been detected on ventriculoarterial shunt catheters, polytetrafluoroethylene peritoneal dialysis fistula, and prosthetic cardiac valves [44]. Although studies establishing the presence of bacterial biofilms are common, there remains a scarcity of studies detailing the presence of non-*Candida* priority fungal pathogens as biofilms on medical devices [31]. Several fungal species of filamentous, yeast, and dimorphic fungi are known to produce polymicrobial fungal biofilms [31]. Removal of the medical device is required to eliminate the fungal infection. The invasive nature of fungal cells owing to the presence of varied morphological forms, yeasts, hyphae, and pseudo-hyphae, however, means that dissemination and subsequent BSIs can occur, leading to prolonged morbidity and often mortality [7]. Biofilms consisting of both bacterial and fungal species together are an important clinical consideration. For example, studies have shown that sessile *C. albicans* hyphae protect sessile *S. aureus* cells, with *S. aureus* also displaying increased resistance to vancomycin and daptomycin [45].

## 3. Current Sterilization Methods

Sterilization requires the complete elimination of microbial life, including spores, by chemical or physical means and is achieved via several methods. Methods include autoclaving with heat/pressure, vaporized hydrogen peroxide (VH_2_O_2_), radiation, i.e., Gamma, E beam, and X-ray, and ethylene oxide (EtO) gas, among others (Table 1). Sterility is defined as the expectation that less than 1 in 1 million devices harbors bacterial spores and is referred to as the sterility assurance level (SAL), i.e., SAL of 10^−6^ [9]. Sterility is achieved when the number of test bacterial spores (*Bacillus atrophaeus*) reaches a SAL of 10^−6^ or less [52]. Disinfection, however, is defined as the elimination of pathogenic organisms from inanimate objects except microbial spores and is further categorized into high-level and low-level disinfection [53]. Cleaning of devices is a prerequisite of sterilization and high-level disinfection [7]. The European Medical Devices Regulation 2017/745 (MDR) and the In Vitro Diagnostic Medical Devices Regulation 2017/746 (IVDR) [8] outline requirements for the control of contamination, infection, and sterility of medical and diagnostic devices. The ISO 22421:2021 [8] sterilization of healthcare products details the requirements for the terminal sterilization of medical devices in healthcare facilities. EN 556-1 [8] is the European standard specifying requirements for designating a terminally sterilized device as sterile [8]. Critical and semi-critical medical devices must be sterilized before use to prevent the risk of infection in end users. Currently, EtO and radiation modalities are the main terminal sterilization methods in use, with thermal (heat) sterilization applied in hospital settings [9]. Device factors, including device complexity, material type, presence of biologics (combination devices), and heat sensitivity, impact the sterilization method applied. Issues relating to carcinogenic emissions from EtO gas have motivated the EPA to propose limiting emissions by 80% or to seek alternative sterilization methods for medical devices. ISO 22441:2022 [8] supports the sterilization of healthcare products using low-temperature VH_2_O_2_ as an alternative to EtO [54]. VH_2_O_2_ offers a viable alternative to EtO as an effective sterilizing agent, which is nontoxic, fast, and leaves no harmful residuals on the product [55]. VH_2_O_2_ sterilizing equipment is implemented in hospital settings as a low-temperature sterilization technology [55]. FDA-approved chemicals used as high-level disinfectants and sterilizing solutions include ≥2.4% glutaraldehyde, 0.55% ortho-phthalaldehyde (OPA), 0.95% glutaraldehyde with 1.64% phenol/phenate, 7.35% H_2_O_2_ with 0.23% peracetic acid, 1.0% H_2_O_2_ with 0.08% peracetic acid, and 7.5% H_2_O_2_ [56]. Non-FDA-approved disinfectants are not used for reprocessing reusable medical devices. These include iodophors, chlorine solutions, alcohols, quaternary ammonium compounds (QACs), and phenolics [56].

### Factors Leading to Sterilization Failure

Lack of compliance with sterilization and disinfection protocol is the main causative factor for device-associated HAIs. Improperly sterilized critical devices containing MDR species have been associated with endoscopy procedures [50]. MDR pathogens, including CRE, were isolated from patients who had undergone endoscopic retrograde cholangiopancreatography procedures using reprocessed duodenoscopes [60]. Research has determined that the internal compartment of intestinal endoscopes can harbor up to 10 log_10_ enteric microbial cells, a significant bioburden [53], with narrow channels, bends, and varied surfaces representing a challenge for cleaning and sterilization. Importantly, the FDA guidance documents do not include specific requirements for the elimination of MDR pathogens from medical devices. Emerging and re-emerging pathogens must also be considered [53]. MDR pathogens display increased biocidal resistance and produce biofilms with increased resistance to disinfection methods. Cleaning is an absolute prerequisite to high-level disinfection and sterilization, where the efficacy of the sterilant is affected by organic matter and organic load present on the device, as commonly seen with endoscopy equipment. Contaminated reusable devices can harbor bacterial and fungal species in planktonic, biofilm, and spore forms that can tolerate reprocessing at varying exposure parameters. Additionally, the complex structure of such devices proves challenging for optimizing uniform exposure to the sterilizing or disinfecting agent [58]. Combination products, medical devices with a biological component (API devices), and novel drug-delivery platforms are increasingly common and represent a difficult-to-sterilize device [9]. Environmental contamination and transmission are important considerations as pathogens transmit from hospital environments to personnel, devices, and patients. Environmental cleaning is therefore essential where the presence of VRE, MRSA, and *Clostridium difficile* have been reduced with effective cleaning strategies [53]. Disinfectants used for cleaning warrant efficacy testing against MDR species as studies have shown elevated levels of biocidal resistance in these species, e.g., QAC resistance in *P. aeruginosa* with resistance to ciprofloxacin [61]. The presence of prions and endotoxins, e.g., LPS toxins, as biological contaminants is another challenge to sterilization methodologies. Prions are more resistant to sterilization than bacterial spores. Therefore, the SAL sterility testing does not guarantee the removal of prions from treated devices [62]. The findings of Sakudo et al. (2022) show that a vaporized mix of hydrogen peroxide and peracetic acid (VHPPA) inactivates prions as determined in mice infectivity assays [62]. Studies have demonstrated the increased efficacy of this combination of oxidizing agents as bactericidal and sporicidal solutions [63]. The heat-stable LPS toxin is extremely resistant to standard sterilization techniques where severe incineration protocols are applied to ensure LPS removal. As such, the prevention and detection of LPS toxin is essential. The Limulus Amebocyte Lysate (LAL)-based assay is universally applied to detect LPS toxin in water supplies. LAL, however, has its limitations [64]. Novel sterilization methods include plasma technology (e.g., hydrogen peroxide gas plasma system), where an electric field is applied to gas plasma, producing UV photons and free radicals with antimicrobial and sporicidal action [65]. Application of low-temperature H_2_O_2_ plasma is limited by its low permeability and the possibility of H_2_O_2_ residual on sterilized devices [59]. Additionally, plasma technology is applicable to surface sterilization only, as it is impacted by the channels and narrow lumens present in endoscopy devices [66].

## 4. Recent Approaches toward Mitigating HAIs

Additional methods of preventing medical device-associated HAIs include the application of surface coatings, incorporating antimicrobial drugs, designing biomaterials to resist biofilm formation, and altering the chemical composition of the device. Incorporation or immobilization of antimicrobial drugs such as antibiotics onto biomaterials does not mitigate the issues associated with resistant species and is not aligned with the Sustainable Development Goals (SDGs) relating to reduced antibiotic use to safeguard public health [67]. The SDGs recognize that the proliferation of AMR can compromise the achievement of the SDGs, affecting health security, poverty, economic growth, and food security. As such, the reduction of AMR is a vital step globally in aligning with the SDGs [67]. Altering device composition and design may impact biocompatibility, functionality, and/or sterilization regimes. Non-antibiotic prophylactic approaches preventing medical device-associated HAIs are desirable and will reduce disease burden and the rate of AMR emergence. Promising biocontrol options demonstrating activity against ESKAPE pathogens and priority fungal species include antimicrobial peptides, bacteriophages, and phage enzymes or endolysins [68]. Biocontrol agents as anti-fouling coatings may prevent biomaterial biofouling (accumulation of micro-organisms on devices) by antimicrobial activity and anti-adhesion action [69].

### 4.1. Antimicrobial Peptides Have Potent Antimicrobial Activity

The application of antimicrobial peptides (AMPs) as antibacterial agents against AMR pathogens shows great promise. AMPs are positively charged (cationic) peptides averaging ca. 30 amino acids in length [15]. Anionic AMPs containing acidic amino acids, e.g., glutamic acid, also exist [70]. AMPs are components of the innate immune system of living organisms with broad-spectrum antimicrobial activity irrespective of AMR and MDR species profiles [71]. As innate immune components, AMPs have anti-inflammatory, anti-cancer, and tissue regeneration properties [72], implying excellent potential as biomaterial components. The cationic nature of AMPs allows for specificity toward the anionic prokaryote cell membrane via electrostatic interactions, where human cell membranes have a net neutral charge [73]. Antifungal peptides are typically short, cationic, and amphipathic. Examples of fungal-specific AMPs include nikkomycins, plant defensins (NaD1), and the echinocandin family of cyclic lipopeptides, which inhibit ergosterol in the fungal cell wall [74]. Bacterial-produced AMPs, termed bacteriocins, have been approved as food bio-preservatives by the FDA, with nisin applied for the control of foodborne *C. botulinum* spores and growth of *L. monocytogenes* in refrigerated dairy [75]. The frog-skin AMP esculentin-1a has demonstrated activity against *P. aeruginosa*, among other strains, and has enhanced the activity of the antibiotic aztreonam [76]. The AMP lactoferrin enhanced the efficacy of fluconazole against fungal species [74]. AMP efficacy relates to a multi-faceted approach resulting from cell lysis, inhibition of macromolecular synthesis, DNA damage, enzyme inhibition, and inhibition of DNA, protein, and nucleic synthesis [15]. This multi-hit approach allows for high potency while limiting the emergence of AMP resistance [77]. Comprehensive reviews of AMP categories, mode of action, and design are provided elsewhere [15,70]. The application of AMPs, including animal-derived polypeptides in vivo, is hindered due to issues including low solubility, cytotoxicity, hemolytic activity, limited activity post-administration, binding to plasma proteins, and degradation, limiting their application therapeutically [70]. Additional factors such as large-scale production, purification, and formulation issues also impact their application (Table 2) [78]. Synthetic AMPs designed from natural AMPs with modifications including adding cationic residues, D-amino acids, cyclization, acetylation, and peptidomimetics are produced to overcome the limitations of natural AMPs while having increased potency [79]. The synthetic AMP AamAP1-Lysine has potent activity against ESKAPE pathogens *P. aeruginosa*, *E. faecalis*, *K. pneumonia*, *and S. aureus* at 5–7.5 μM with the synthetic AMP Guavanin 2 having additional activity against *E. coli*, *A. baumannii*, *C. albicans*, and *C. parapsilosis at* 6.25–20 μM, significantly lower than the natural unmodified AMPs [77]. To prevent human cell toxicity and reduce degradation in vivo, post-translational modifications (PTMs), e.g., pegylation and amidation, can be applied [15]. Synthetic AMP peptoids as AMP mimics have demonstrated self-assembly and excellent antimicrobial activity while resisting proteolytic digestion [80].

#### Antimicrobial Peptides as Surface Coatings

Immobilization of AMPs to biomaterial surfaces is achieved via physical methods (adsorption), chemical methods (via covalent bonding or as Self-Assembled Monolayers (SAMs)) or substrates including gold, titanium dioxide, silicone, and polymer resins [83]. Covalent immobilization of AMPs on material surfaces can allow for microbial contact killing to prevent adhesion and biofilm formation while having improved AMP stability and biocompatibility [82]. As such, research has focused on covalently immobilized AMPs [69], where studies show that covalent immobilization is more efficient than physical adsorption [84]. Natural AMPs, including human lactoferrin-derived peptide hLf1-11 and the human cathelicidin LL-37, have been immobilized on biomaterial surfaces [85]. Importantly, LL-37 can neutralize the LPS toxin of Gram-negative pathogens, reducing inflammation as determined in rat models [86]. The AMP melimine has been covalently coated onto contact lens surfaces for the prevention of eye infection [87]. The derivative Mel4 displayed antimicrobial activity and no eye toxicity in test rabbits in vivo for up to 7 days [87]. Studies describe the increased anti-adhesion activity of AMP bounded chitosan, compared to free chitosan in film coatings against Gram-positive bacterial species [88]. Research on an immobilized peptoid AMP mimic, with a polymer tether against pathogens, including *P. aeruginosa* and *S. aureus*, demonstrated surface antimicrobial activity and reduced bacterial attachment [85]. Self-assembling peptoids, e.g., TM1, demonstrate increased antimicrobial and antibiofilm activity against the ESKAPE pathogens over non-assembling variations [68]. The application of an antimicrobial AMP hydrogel for dermal application containing the AMP RRP9W4N had broad-spectrum activity, increased stability, and biocompatibility with fibroblast cells [82]. The coating or immobilization of medical device/biomaterial surfaces with AMPs has demonstrated anti-adhesion, anti-colonization, and antibiofilm formation action [81]. Furthermore, immobilization of AMPs has numerous advantages in vivo, including reduced cytotoxicity, enhanced stability, and prevention of protein binding [88]. The studies of Nielsen et al., 2022 describe the antibacterial and antibiofilm activity of self-assembling AMP peptoids against ESKAPE pathogens [80]. Contact between an immobilized AMP and microbial species is a consideration for surface coatings, where contact with cell membranes is typically required for inactivation. The efficacy and mode of action of immobilized AMPs differ from that of soluble AMPs and are influenced by exposure, density, orientation, immobilization technique, and type of biomaterial [88]. A range of solid surfaces have been successfully modified with AMPs, including polymers, titanium, gold, stainless steel, and glass [69].

### 4.2. Bacteriophage Antimicrobial Agents

Bacteriophages (phages) are self-replicating viruses that infect bacterial species with prominent levels of specificity while having good biocompatibility, making them ideal candidates for the control of infectious diseases (Table 3). Mycoviruses, which selectively infect fungal species, have demonstrated efficacy against clinically relevant *Aspergillus fumigatus* and *Candida* [89]. Studies have demonstrated the antibacterial efficacy of phages against ESKAPE pathogens for the treatment of gastroenteritis, sepsis, and wound infections [90]. Phage therapy against *A. baumannii* proved effective in treated mice with higher rates of survival [91]. The Enterococcal lysin PlyV12 has activity against *Streptococcus*, *Staphylococcus*, *Enterococcus faecium*, and *E. faecalis* [92]. Clinical trials assessing the intravenous (IV) treatment of *S. aureus* demonstrated excellent results with good phage tolerance in patients [93]. The FDA has approved phage treatment for illnesses including infections of prosthetic joints, bone and implant infections, wound infections, diabetic foot infections, and acute tonsillitis [94]. Phage treatment for *S. aureus* prosthetic valve endocarditis provided noticeable clinical improvement in critically ill patients [95]. Mycoviruses and bacteriophages have the potential for clinical use. However, the administration is hindered by their limitations, including lack of clinical data, thermal instability, formulation and administration difficulties, phage-induced release of endotoxins, and transmission of virulence genes [90]. More recent studies investigate the application of phages as self-assembling scaffolds and biomaterials due to their unique morphology [96]. Liposome and polymer-encapsulating phages are designed for improved delivery stability [94]. Methods to improve phage stability include spray-drying, freeze-drying, emulsion, and polymerization methods. Stability, however, remains variable for gels, powders, and liquids [97]. The incorporation of phages into wound dressings can impart targeted antibacterial activity for disease prevention. Such bioactive biomaterials have been investigated against *P. aeruginosa* and *E. coli* [98]. There are many considerations for the immobilization of phages onto medical devices, including phage orientation, access to pathogen recognition receptors, phage density, and phage shedding [99]. The immobilization of phages into formulations, including powders, creams, gels, and dressings, may offer essential disease treatment options for applications such as medical device coatings, phage enzymes (endolysins) may offer a more applicable approach due to their proteinaceous structure.

#### Antibacterial Endolysins

Phages produce enzyme endolysins (lysins) and virion-associated peptidoglycan hydrolases, which pierce the bacterial peptidoglycan layer, enabling entry into the cells [90]. The cell membrane of Gram-negative bacteria acts as a barrier to lysin activity, limiting antibacterial efficacy [101]. Endolysins have many advantages as antibacterial agents, including bactericidal specificity, rapid cell lysis, anti-biofilm action, lack of resistance mechanisms, and synergism with antibiotics [100]. Studies describe the efficacy of endolysins against MDR ESKAPE pathogens *A. baumannii*, *E. coli*, *K. pneumoniae*, *P. aeruginosa*, and MRSA [102]. Studies have also determined the potency of lysins LysSS and CHAP-161 against catheter-contaminating bacterial species with a MIC range of 250 to 750 mg/mL [103]. Endolysins, including LysH5 and phi11, successfully eliminated biofilms of *S. aureus* and destabilized sessile cells within the biofilm matrix [92]. The endolysin LysECD7 reduced the biofilm-forming capacity of *K. pneumonia* and degraded formed biofilms in vitro [104]. The endolysin LysPA26 produced a ca. 2 log reduction of *P. aeruginosa* biofilms on polystyrene surfaces [100]. Staphefekt SA.100 is an endolysin-based product (cream or gel) marketed in the EU for dermal application with long-term activity against *S. aureus* [105]. Endolysin PlySs2 provided a 99% reduction of planktonic cells and a 75% reduction in biofilm formation compared to vancomycin [95]. P128 (NCT01746654) and N-Rephasin^®^ SAL200 (NCT03089697) are endolysin-based therapeutics currently in phases II and IIa clinical trials for the treatment of *S. aureus* in vivo, respectively [101]. Lysins are susceptible to protein degradation and have poor oral bioavailability. The immobilization of endolysins on nanoparticles has enhanced protein stability and efficacy [100].

## 5. Considerations in the Application of AMPs, Phages and Endolysins

There are important hurdles to overcome to effectively apply these biological agents as antimicrobial approaches against MDR species [106]. A provision of large volumes is required to formulate enough products for clinical application. Bioprocessing using suitable expression systems offers a means of biological production. The application of recombinant DNA technology and genetically engineering cell expression systems, including bacterial, yeast, fungi, plant, and animal, for the large-scale production of AMPs, phages, and lysins offers a potential platform for production. Bioprocessing in continuous mode bioreactors offers advantages over batch systems, including the addition of fresh media and removal of product in a continuous stream, higher yield, lower cost, and easier control [107]. *E. coli*, for example, has been applied to produce endolysins active against *S. aureus* [101]. Issues arise, however, with phage and endolysin toxicity to the bacterial expression system, aggregation of proteins, and the production of LPS toxins by *E. coli* [15]. The yeast *Pichia pastoris* offers another system that gives a high protein yield at a low cost in the absence of LPS while not being susceptible to lysin toxicity [108]. Yeast and other eukaryotic expression systems possess the ability to perform post-translational modifications, which increases biocompatibility and efficacy in vivo [109]. As proteinaceous materials, the potential of immunogenicity and loss of stability in vivo of AMPs and lysins must be determined. Pharmacokinetic and pharmacodynamic profiles must be established to determine the route of administration and bioavailability in vivo. Orally administered peptides and proteins are prone to enzymatic degradation in the GIT, while IV-administered therapeutics may bind to circulating blood proteins [15]. The fermentation process for phage production is limited by phage and bacterial mutation rates and host cell death within the bioreactor [107]. The bacterial expression system may become resistant to phage infection, or the phage structure may change from the desired structure [110]. The bacterial nucleic acid and protein content post-cell lysis impacts the downstream processing of phages and peptides from bioreactors and large-scale production [111]. The presence of bacterial nucleic acids, proteins, enzymes, and LPS toxins in the final product impacts biocompatibility [111]. For IV phage formulation, such impurities are not acceptable, particularly LPS toxin, which is associated with fever, leukopenia, leukocytosis, and fatal sepsis [107]. Additionally, downstream processing, e.g., purification methods, can impact the purity of the phage product [110]. For clinical use, the stability of formulated products is essential for regulatory approval, and cold storage and shelf life are important considerations [97]. Issues relating to the leaching of materials from the medical device, release kinetics, and the amount of coating applied must be considered. The leaching of non-biocompatible compounds from devices hinders their application in vivo. The use of biocompatible AMP and phages may offer suitable options. Investigative studies, however, must determine optimal coating conditions. Sterilization of thermosensitive protein products remains a challenge for AMP, endolysin, and phage biomaterials or therapeutics. Proteinaceous material is heat-sensitive and is not compatible with steam or heat sterilization methods. EtO gas and Gamma irradiation methods may offer a suitable sterilization approach to such biomaterials. Studies, however, have assessed the loss of phage viability following exposure to EtO and Gamma sterilization in formulated powders [112]. Studies assessing the impact of radiation on AMPs determined that the shape of certain AMPs affected their solubility, with linear AMPs holding their antibacterial activity [113]. VH_2_O_2_ has negative effects on protein therapeutics, including oxidation and aggregation [114], which may lead to product instability. Successful sterilization of such biomaterials or therapeutics represents a significant challenge limiting their application clinically. Research is warranted to determine effective modalities with post-sterilization biocompatibility and efficacy testing. Additionally, obtaining regulatory compliance for novel sterilization methods limits their application for medical devices. A lack of global compliance with sterilization procedures and protocols is a major bottleneck to sterilization using novel devices [115]. Harmonization of sterilization methods is part of the MDR, where conformity in testing and sterilization methods is sought. Currently, the FDA has no standards in place for novel methods, including pulsed light, H_2_O_2_, and ozone [50].

## 6. Conclusions

Significant in vivo research is warranted to fully establish the pharmacokinetic and pharmacodynamic profile of phages, lysins, and AMPS as standalone or combination products. In vitro efficacy studies demonstrate promising results. However, the influence of biological systems on formulated end products must be established. AMPs, phages, and lysins demonstrate antibiofilm activity, supporting their application as medical device coatings. Phages and phage cocktails have displayed potent antibacterial activity against MDR ESKAPE pathogens. Combinations of phages and AMPs can be applied to enhance selectivity and potency. The application of RDNA technology and cell culture systems for expressing heterologous proteins has become a valuable tool in the production of biologics. The hurdles of large-scale production must be overcome by optimizing expression systems to provide a high yield of biologics with optimal PTMs and reduced impurities. Downstream processing, purification, and formulation issues to ensure stability with storage and transport are crucial factors. Biomaterials, medical devices, and therapeutics formulated with AMPs, lysins, or phages suffer the same restrictions as other biologics in terms of sterilization options. Regulatory frameworks outlining the regulatory requirements of phages, phage cocktails, lysins, and AMPs need to be established to provide uniform testing and formulation procedures to allow for comparative testing in vitro and in vivo. Although significant advances are being made in the clinical use of these antibacterial agents, the ability to produce, formulate, and sterilize end products hinders their application.

## Figures and Tables

**Table 1 ijms-25-00201-t001:** Sterilization methods for medical devices, advantages, disadvantages, and limitations.

Sterilization Method	Advantages	Disadvantages	Limitations	Regulatory Standards *
		Traditional Methods		
Most heat (steam)	Nontoxic, quick, penetrates packaging, less affected by contaminants than other methods [53]	Not effective against endotoxins, can soften, degrade, and hydrolyze polymer-based materials, presence of water on devices [53]	Not used for heat-sensitive items	ISO 17665-1:2006(en), CEN/ISO TS 17665-2, EN ISO 11138-3 [8,54]
Dry heat	Convenient method for devices/components that are not temperature-sensitive	Increased time for sterilization, damage-sensitive materials,	Not suitable for plastic and rubbers and heat-sensitive material	EN ISO 20857, EN ISO 11138-4 [54]
Radiation— E Beam, Gamma, X-ray	Reduced processing time, good penetration capability, and flexibility to process different dose ranges and densities within the same process run	Possible lethal exposure to radiation and ozone [9], production of ozone gas, not as suitable to products containing biologics or tissues, cobalt supply impacts Gamma radiation	Certain viruses, e.g., HIV-1 and bacteria (*Streptococcus faecium* and *Micrococcus* species) are more resistant to irradiation [9], distance between devices and radiation is a limitation of E beam	ISO 11137-1:2006/Amd 2:2018, ISO 11137-2:2013/Amd 1:2022 [8,54,57]
Ethylene oxide	Does not damage the device, suitable for a range of material types, EO can sterilize moisture- and radiation-sensitive materials	Residual ethylene oxide gas, raises health concerns, EO is toxic, carcinogenic, and explosive, lengthy processing and aeration time	EO sterilization must be conducted in a humid environment.	ISO 11135:2014 [8] Residuals governed by ISO 10993-7:2008(R)2012 [57] EN ISO 11138-2 [8]
VH_2_O_2_	Nontoxic cold gas sterilant, short processing exposure times, low temperature, excellent material compatibility, green breakdown products, permeates most materials, inactivates prions [52]	Poor penetration properties, terminal sterilization only, high initial costs, damages nylon	Some material incompatibility and vapor penetration may be limited, not suitable for liquids [53]. Irritating to skin, eyes, and respiratory system [55]	ISO NP 22441, ISO 11138-6 [54,57]
**Novel methods**				
Chlorine dioxide gas	Good material compatibility [58]	Pre-humidification of ClO_2_ is mandatory, corrosive [58]	Poor penetration, user safety issues,	Standards are not in place for implementation
Vaporized peracetic acid (VPA),	Sterilizes at room temperature, no harmful chemical residuals, sporicidal, lower processing time than EtO [52]	Poor compatibility with materials such as lead, brass etc., potential for dermal and eye toxicity [52,59]	Limited clinical experience in using VPA, not widely suitable for terminal sterilization of combination products	
Nitrogen dioxide	Low temperatures, flexible cycle conditions, compatibility with most polymers and stainless steel, broad-spectrum antimicrobial [58]	NO_2_ is a surface sterilant, incompatible with PU, nylon, and copper [58]	Surface sterilization only	
Low-temperature gas plasma	Low risks, Nontoxic, potent antimicrobial, versatile application [59]	Low permeability of H_2_O_2_ [59]	Not suitable for complex shapes and narrow lumens	
UV, Pulsed light	Non-thermal technologies, no residuals on device, sporicidal pulsed light [32]	Adaptive molecular and cellular repair mechanisms lead to recovery of viability [32]	Poor penetration, user safety issues, materials like glass and plastics absorb UV irradiation [32]	Pulsed light is FDA-approved for food application [32]

* Additional relevant standards for sterilization EN ISO 13408 [8] presents general requirements of aseptic processing, EN ISO 11737-1 [8] Microbiological methods, and EN ISO 11138 Biological indicators [54,57].

**Table 2 ijms-25-00201-t002:** Considerations on the application of AMPs as antimicrobial medical device coatings.

Advantages	Considerations
Broad-spectrum antimicrobial activity against yeast, fungi, viruses, and bacteria [15]	Biocompatibility issues to the patient [15]
Effective against AMR/MDR species [71,74]	Limited information on in vivo antimicrobial efficacy [15]
Prevent excessive activation of pro-inflammatory by LPS toxin [81]	Bacterial resistance may emerge to certain AMPs, activity of efflux pumps [74]
Potent action—low concentration needed [80]	Limited stability and short half-life [82]
Antibiofilm activity was observed in some cationic AMPs [74]	Protein and enzymatic degradation in vivo [70]
Rapid onset of action [15]	Difficulties in producing large quantities of AMPs [15]
pH stability—varying pH ranges throughout the GIT [15]	Overstimulation of the immune system may be an issue—pro-inflammatory cytokines [80]
May boost the immune system of the patient [74]	Influence of in vivo conditions on AMP activity needs investigating [15]
Can be modified synthetically and post-translationally [15]	Impact of sterilization modalities on AMPs needs investigating
Self-assembly in a physiological environment [80]	Low solubility [15]

**Table 3 ijms-25-00201-t003:** Considerations on the application of bacteriophages and endolysins as antimicrobial medical device coatings.

Advantages of Phage’s	Advantages of Endolysins	Considerations
Highly species-specific [67]	No resistant bacteria evident to date [90]	Thermostability issues [94]
Highly potent [90]	Enzymes have a broader range of specificity	Large-scale production issues [90]
Use of phage cocktails for polymicrobial applications [96]	No risk of transferring virulence genes [67]	Resistance to phages is possible [96]
Antibiofilm activity [67]	Penetration of biofilm matrix [100]	Endolysin enzyme saturation kinetics [90]
Self-replicating requiring low doses [90]	Relatively fast-acting [100]	May need outer membrane destabilizers present, which may be toxic [90]
Self-limiting once the bacterial pathogen is eliminated [100]	Synergy with antibiotics is possible [67]	Endolysins are not self-replicating like phages
Not FDA-approved [67]		May be influenced by environmental factors [92]
Can be used in conjunction with other biocontrol measures [90]		Phages can carry and transmit toxin and virulence genes [96]
Biocompatible [90]		Shelf life, storage issues [90]
Prokaryote specific [67]		No LPS activity [101,102]
Not impacted by AMR/MDR phenotypes [100]		
Low risk of hypersensitivity or allergic reactions [100]		

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
