# Peer review of "Medical Device-Associated Healthcare Infections: Sterilization and the Potential of Novel Biological Approaches to Ensure Patient Safety"

_ijms, 2023, doi:10.3390/ijms25010201_

Round 1

Reviewer 1 Report

Comments and Suggestions for Authors

The authors present an comprehensive review on current sterilization strategies apart from covering problem of HAIs in clinics.

I have following suggestions to make:

(a) The authors shall highlight the problems associated with coating solutions i.e. leaching out, limited reservoir or inconsistent release kinetics and possible solutions to overcome.

(b) As disinfection and sterilization are different processes and are applied on different types of medical devices and surgical instruments, the authors shall bring about a segregated discussion on sterilization and disinfection methods with their respective application in specific devices. Else, the authors may remove disinfection from title and/or abstract.

(c) As the authors give an elucidated view on sterilization methods, it shall be of great interest to audience to know the bottlenecks for obtaining regulatory compliances with newer sterilization methods.

(d) It is necessary to briefly discuss in section 5 about the absence of sterilization procedures in research efforts that develop novel coating solutions.

Comments on the Quality of English Language

A moderate editing of English shall be required

Reviewer 2 Report

Comments and Suggestions for Authors  

This review addresses the critical factors contributing to healthcare-associated infections (HAIs), encompassing issues such as the inadequacies in sterilization and disinfection of reusable devices, microbial virulence factors that facilitate persistence on devices, and the repercussions of WHO priority pathogens on morbidity and mortality rates among affected patients. The review also explores the feasibility of applying novel antimicrobial techniques in clinical settings as part of antimicrobial stewardship efforts.

Comments:

1. The manuscript introduced the bacteriophages, endolysins and antimicrobial peptides, but there are lots of new health devices with metal ion, cationic ion and other antimicrobial chemicals coating proved by FDA every year. Why didn't discuss these new techniques? If author wants to focus on bacteriophages, endolysins and antimicrobial peptides only, I suggest using a more specific title.

2. Line 309-312, why did author stated "Incorporation or immobilization antimicrobial drugs on to biomaterials does not mitigate the issues associated with resistant species", but there already many reports for other new antimicrobial coated-devices which could effectively inhibit drug-resistant pathogens? Can author also explain the details in SDGs about antibiotics?

3. Table 1, please cite the original research for each novel method. And there should be a sub-title like "conventional methods" under the first row.

4. Table 2 and 3, please provide reference for each advantage and consideration in the tables.

5. Is there any medical devices products using bacteriophage or endolysins coating?

Round 2

Reviewer 1 Report

Comments and Suggestions for Authors

Please attend to line no. 507 to 509 as it is not clear. The manuscript in its present form can be accepted as it has incorporated suggestions made in previous report.

Comments on the Quality of English Language

The English of the manuscript needs minor corrections.

Author Response

Line 507 has been rewritten. grammar has been checked throughout.

thank you very much
